# Localization of Colorectal Cancer Lesions in Contrast-Computed Tomography Images via a Deep Learning Approach

**DOI:** 10.3390/bioengineering10080972

**Published:** 2023-08-17

**Authors:** Prasan Kumar Sahoo, Pushpanjali Gupta, Ying-Chieh Lai, Sum-Fu Chiang, Jeng-Fu You, Djeane Debora Onthoni, Yih-Jong Chern

**Affiliations:** 1Department of Computer Science and Information Engineering, Chang Gung University, Guishan, Taoyuan 33302, Taiwan; pksahoo@mail.cgu.edu.tw (P.K.S.); d0521006@cgu.edu.tw (P.G.); d0421008@cgu.edu.tw (D.D.O.); 2Department of Neurology, Chang Gung Memorial Hospital, Linkou, New Taipei City 33305, Taiwan; 3Department of Medical Imaging and Intervention, Chang Gung Memorial Hospital, Linkou, New Taipei City 33305, Taiwan; cappolya@cgu.edu.tw; 4Department of Metabolomics Core Lab, Chang Gung Memorial Hospital, Linkou, New Taipei City 33305, Taiwan; 5Division of Colon and Rectal Surgery, Chang Gung Memorial Hospital, Linkou, New Taipei City 33305, Taiwan; sumfu@cgu.edu.tw (S.-F.C.); you3368@cloud.cgmh.org.tw (J.-F.Y.); 6College of Medicine, Chang Gung University, Guishan, Taoyuan 33302, Taiwan; 7Graduate Institute of Clinical Medical Sciences, College of Medicine, Chang Gung University, Guishan, Taoyuan 33302, Taiwan

**Keywords:** colorectal cancer, deep learning, localization, computed tomography

## Abstract

Abdominal computed tomography (CT) is a frequently used imaging modality for evaluating gastrointestinal diseases. The detection of colorectal cancer is often realized using CT before a more invasive colonoscopy. When a CT exam is performed for indications other than colorectal evaluation, the tortuous structure of the long, tubular colon makes it difficult to analyze the colon carefully and thoroughly. In addition, the sensitivity of CT in detecting colorectal cancer is greatly dependent on the size of the tumor. Missed incidental colon cancers using CT are an emerging problem for clinicians and radiologists; consequently, the automatic localization of lesions in the CT images of unprepared bowels is needed. Therefore, this study used artificial intelligence (AI) to localize colorectal cancer in CT images. We enrolled 190 colorectal cancer patients to obtain 1558 tumor slices annotated by radiologists and colorectal surgeons. The tumor sites were double-confirmed via colonoscopy or other related examinations, including physical examination or image study, and the final tumor sites were obtained from the operation records if available. The localization and training models used were RetinaNet, YOLOv3, and YOLOv8. We achieved an F1 score of 0.97 (±0.002), a mAP of 0.984 when performing slice-wise testing, 0.83 (±0.29) sensitivity, 0.97 (±0.01) specificity, and 0.96 (±0.01) accuracy when performing patient-wise testing using our derived model YOLOv8 with hyperparameter tuning.

## 1. Introduction

Colorectal cancer (CRC) has been the third most commonly diagnosed cancer and the second most deadly cancer in recent years. In 2020, CRC accounted for 10% of the global cancer incidence and 9.4% of cancer deaths. Due to the elevated exposure to environmental risk factors and lifestyles and diets shifting toward westernization, it is predicted that the number of new cases of CRC will reach 3.2 million in 2040 [1]. The estimated number of new CRC cases is highest in China and the United States, where it is estimated that the number of new cases will increase by 64% from 0.56 million in 2020 to 0.91 million in 2040. In the case of the United States, the number of new cases is estimated to increase from 0.16 million in 2020 to 0.21 million in 2040. Cancer remains the top cause of death in Taiwan for the 38th year in a row. CRC is the most common cancer in males and the second most common cancer in females [2]. In recent years, significant attention has been focused on leveraging machine learning and deep learning approaches to enhance clinical practice for patients with CRC [3,4]. Demographic data, patient family histories, and histological factors have been correlated using machine learning algorithms such as multi-layer perceptrons or random forests to predict CRC and survival rates [5]. In addition, deep learning models have been employed to aid in the histopathologic analysis of tissues obtained from patients diagnosed with CRC. Slides containing the tissue samples are digitally scanned and analyzed using deep learning models such as EfficientNet [6] and Inception-ResNet-v2 [7] to identify and differentiate the cancerous regions from normal colonic tissues in the scanned slides [8].

Abdominal computed tomography (CT) is a frequently used imaging modality for evaluating gastrointestinal diseases. The localization of colorectal cancer is often realized using contrast-enhanced CT (CCT) before a more invasive colonoscopy [9]. CCT also plays an essential role in evaluating the local tumor spread and distant metastatic status, which then dictates the appropriate treatment. Despite the continuous advancement of CT scanners and imaging techniques, clinicians and radiologists still face many challenges when interpreting colorectal disease in CT. Identifying CRC can be challenging in CT scans due to the relatively limited attention given to the colon in studies primarily focused on other diseases [10,11]. The sensitivity of CT in detecting CRC greatly depends on the tumor size. Moreover, small tumors may be obscured by fecal materials within the colon when CT is performed without bowel preparation (i.e., in routine abdominal CT). A study showed that missed and incorrectly diagnosed CRCs in CT had an average size of 3.3 cm and length of 5.1 cm [12]. In addition, the absence of pericolic fat stranding, engorged vessels, and associated lymphadenopathy may result in the misidentification of CRC. If a CT exam is performed for indications other than colorectal evaluation, the entire colon and rectum may not be analyzed carefully and thoroughly. Due to these factors, a significant percentage of CRCs, ranging from 6 to 20% depending on the CT imaging techniques, are undetected in routine abdominal CT, which may lead to delayed diagnoses of CRC [13]. A study demonstrated that early CRCs that are discernible in CT might take as long as 2 years to develop into clinically symptomatic lesions. It is of great survival benefit to correctly identify asymptomatic CRCs in CT. Due to CT’s widespread use in daily clinical practice, missed incidental colon cancers in CT are an emerging problem for clinicians and radiologists, and a wide variety of non-neoplastic colorectal diseases may mimic CRC in CT.

Colonoscopy is used to identify CRC; however, the technique is invasive and can only be performed in patients with bowels prepared before the examination. Consequently, colonoscopy is sometimes ineffective in case of unprepared bowels, difficulty in completing colonoscope insertion, and emergent cases, and CT shows low sensitivity in cases of smaller cancer sizes and unprepared bowels. This is because the symptoms of early CRC are rarely noticed, causing radiologists to pay little attention to every inch of the large intestine. However, if radiologists follow specific rules to observe the colon and rectum, early diagnosis sensitivity can be improved. This is illustrated by the work in [9], where the sensitivity in reviewing CRC increased from 66% to 86% when radiologists followed a diligent search pattern to identify CRCs in the large intestine. Different techniques for observing patterns include the thickening of the bowel wall and segments containing minimal oral contrast.

Although the CT imaging features of CRC have been well established, there is substantial overlap between the imaging presentations of benign and malignant colorectal diseases [14]. Moreover, broad-spectrum colonic complications such as colitis, perforation, or abscess formation can occur in patients with CRC, making it even more challenging to achieve a correct diagnosis [15]. Furthermore, both CRC and diverticular disease are common in elderly patients; thus, both benign and malignant entities can co-exist in a patient. Considering the different types of research performed on CRC detection, it is evident that early detection of CRC is necessary to improve the chances of survival. 

### Motivations

Although short segmental wall thickening with a polypoid or ulcerative appearance is characteristic of CRC, these imaging features may be subtle in CCT for early CRC. The colon is a long, tortuous, and possibly overlapping tubular organ; thus, a dedicated search is usually required to accurately detect CRC in CT. The CCT imaging features of CRC have been well established, but there is substantial overlap between the imaging presentations of benign and malignant colorectal diseases. The accuracy of CT in preoperative cancer detection is unsatisfactory, with a reported accuracy ranging from 48% to 77%. Physicians who meet patients with abdominal CT may not be professional radiologists or colorectal specialists such as colorectal surgeons, so colorectal lesions may be ignored or misdiagnosed as other diseases. This inter-variability and intra-variability in observations may create confusion among physicians, where accuracy depends on the experience and expertise of the physicians. In addition, each CT scan for a patient contains 150 slices, which must be examined manually to check for abnormality, and the whole process is tedious and time-consuming. Moreover, there are several abnormalities with similar appearances, such as diverticulitis, Crohn’s disease, ulcerative colitis, diffuse ischemic colitis, and polyp. Furthermore, the appearance and location of the colon and rectum vary in all the slices. As a result, manual differentiation of the disease is challenging since a CRC lesion often exhibits similar attenuation to the surrounding colon tissue, and there is no specific preferred location for the occurrence of the lesion. Therefore, there is a pressing need for artificial intelligence to automatically detect potential colorectal lesions in daily clinical practice within radiology.

## 2. Related Works

Accurately discriminating CRC from benign entities requires years of experience in CT interpretation, posing a great challenge for non-expert clinicians or radiologists. Several studies have been carried out to design different models for the AI-based analysis of colorectal cancers in CT images. For instance, research that focuses on the early diagnosis of rectal cancer using a machine learning approach is discussed in [16]. Several features such as location, orientation, and scale are extracted and used as inputs to train the artificial neural networks using backpropagation algorithms. For the training procedure, the optimal sets of weights are obtained via adjusting the weights for a backpropagation algorithm hybrid with a genetic algorithm (GA) and a firefly algorithm (FFA). The database consisted of approximately 140 colon images, 59 images of which were used for training, 40 for validation, and the remaining 41 for testing. Out of 120 abnormal CT colon images, 32 CT images belonged to benign polyps in the colon, another set of 32 CT images belonged to adenomatous polyps, the third set of 33 images belonged to moderately malignant polyps, and finally, the remaining 33 CT images belonged to the highly malignant category. Approximately 10 CT images without any defects (normal) were also considered. This categorization was automated and confirmed using the backpropagation algorithm (BPA) in combination with GA and FFA for weight optimization. The sensitivity obtained for benign, adenomatous, and moderately malignant polyp identification was 0.86, and 0.88 for highly malignant polyps. A domain-based analysis of colon polyps was proposed in [17] to improve polyp detection accuracy in the CT images of the abdomen and pelvis. The work proposed effective semi-automatic colon detection, segmentation, removal of tagged fecal matter through electronic cleansing, and measurement of the size of polyps in CT images. The experiments were carried out using 40 CT colonography datasets, where polyp sizes of <10 mm were correctly detected with an accuracy of 95.82% and a standard deviation of 0.68.

In addition, an automatic segmentation of colorectal tissue was performed using deep and hierarchical learning, wherein a CNN was used for the colon tissue feature extraction. Base analysis was performed to allow the classification of each center pixel as either a colon tissue or a background tissue. The CNN architecture took an input patch size of 28*28, justifying that a smaller patch size provided enough information to classify the tissues and therefore achieved higher effectiveness in the performance. The open dataset consisted of approximately 1600 gray-scale slices of CT images obtained from well-prepared colons, which were preprocessed to generate 28*28-sized patches, as required by the CNN model. Ground-truth generation was performed manually via assigning a label of 1 if the center pixel belonged to colon tissue and 0 otherwise. The proposed method in [18] achieved a sensitivity of 96.9%, a specificity of 98.7%, and an accuracy of 97.9%. However, the study used well-prepared CT colon images, which can rarely be obtained in emergent cases. 

Similarly, a 3D segmentation model was proposed for the automatic segmentation of the prostate, bladder, rectum, and femoral heads in male pelvic CT images [19]. The data were collected from the University of Texas Southwestern Medical Center, consisting of 136 patients with metastases. A multichannel 2D U-Net followed by a 3D U-Net with an encoding arm consisting of a residual network variant known as ResNeXt was used as the segmentation model. The proposed model achieved mean dice coefficient score values of 90% with a standard deviation of ±2.0 for the prostate and 84% with a standard deviation of ±3.7 for the rectum. Their proposed work aimed to segment the organs surrounding the prostate cancer to find the risk of radiotherapy treatment planning. Although the works discussed above proposed the detection and segmentation of colorectal regions and lesions in the CT scan images of well-prepared bowels, it is, nonetheless, difficult to locate colorectal lesions in emergent cases due to the presence of feces and other abnormalities in the long, tortuous, and tubular structure.

## 3. Materials and Methods

### 3.1. Dataset Acquisition

Retrospective CT image data were collected according to the Institutional Review Board approved by Chang Gung Memorial Hospital, Linkou, Taiwan, and the need for patient informed consent was waived. The images were collected in the Digital Imaging and Communications in Medicine (DICOM) format with anonymized information. The CT data used in this study were collected from January to December 2020, downloaded from the picture archiving and communications system (PACS) in Chang Gung Memorial Hospital. A total of 232 patients were included in this study. The inclusion criteria were as follows: Phase 1 of the data acquisition consisted of 143 patients with 38,986 slices with both non-contrast and contrast CT. However, based on the scope of our study, we included 19,799 slices of contrast CT in the axial plane (C+ axial/CCT) for our analysis. All the lesions were confirmed as CRC via pathological analyses or a combination of the typical image performance and clinical data. This was followed by phase 2 of the data collection with data collected from 102 patients. However, based on our inclusion criteria, 13 patients were excluded from our study. Therefore, phase 2 of the data collection consisted of 89 patients with 10,859 C+ CT slices, 661 of which contained a CRC region. We used 143 patients’ data for the model’s initial training, followed by improving the model performance with the addition of 47 patients’ data, which is a practical incremental training approach wherein the data collection and model training are performed simultaneously. The final derived model was evaluated with the remaining 42 patients’ data out of the 89 patients collected in phase 2 of the data collection. 

The CT exams were conducted using a 64-detector-row CT scanner (Aquilion 64; Toshiba Medical Systems, Otawara, Japan). An enteric contrast medium was not administered, but an intravenous contrast medium consisting of 100 mL of iohexol (350 mg of iodine per milliliter, Omnipaque 350; GE Healthcare, Princeton, NJ, USA) was administered via a power injector at a rate of 3 ml/s. The institutional standard CT protocol for CRC patients involves contrast-enhanced imaging during the portal venous phase, with a delay of 70 seconds after injection. All CT scans were conducted from the chest to the pelvis. The CT scan parameters used were as follows: 120 kVp and automatic tube current modulation. The images were reconstructed to a thickness of 5 mm with a 5 mm interval.

### 3.2. Ground Truth Annotation

In the patients’ CT scans, the tumors were observed to be localized either in the colon or rectal regions. Irrespective of the location of the CRC, it was necessary to annotate the CRC region for training the deep learning-based detection model in a supervised manner. As a result, two radiologists with approximately 5 years of experience annotated the CRC region in the raw CCT slices using OSIRIX MD v10.0.5 (Pixmeo SARL, Bernex, Switzerland) [20] and Horos (https://horosproject.org/) (version 3.3.6) (accessed on 20 June 2021). Figure 1 shows examples of the ground truths. A total of 1558 slices were annotated out of 16184 slices in the 190 patients used for the model derivation.

### 3.3. Data Preprocessing

#### 3.3.1. Slice Selection

Several substeps were involved in the preprocessing stage. Not all the slices from the images collected for processing and analysis were required for our analysis. For instance, the CT exams incorporated images from the chest to the pelvic regions with multiplanar reformation. Only axial images from the abdomen to the pelvis were used for our analysis. In addition, when the CT scan images were obtained in the DICOM format, the radiologists needed to manually select the appropriate slices based on the appearance of the colon and rectum. Slices were excluded if they did not contain any portion of the colon or rectum. Therefore, the suitable slices were manually selected by the experts based on the inclusion and exclusion criteria.

#### 3.3.2. Slice Conversion

The suitable slices were selected and afterward converted from the DICOM format to the JPG format using the open-source DICOM viewer, Horos software (https://www.horosproject.org/) (version 3.3.6) (accessed on 20 June 2021). Using this software, all slices could be exported in a particular series to the desired format, which was JPG in our case. As shown in Figure 2, we exported the DICOM slices belonging to the C+ axial series in the JPG format based on our requirements.

#### 3.3.3. Automatic Cropping and Resizing

As shown in the input in Figure 3, the CT image obtained contained a black back-ground that was not required for analysis. Therefore, we used the automatic cropping method in Python to remove the unwanted region. Finally, the cropped images were automatically resized to 512*512. 

#### 3.3.4. Bounding Box Labeling

We obtained ground-truth information from the doctors’ annotations of the ROIs using Horos. However, to automatize the colorectal lesion localization method using AI, the machine must be trained to learn and make predictions based on correlations. This training can be achieved via collecting and feeding the data to the training models. However, in the case of supervised learning-based deep learning analysis, it is essential to annotate the data to provide labels for the machine’s training. As a result, we used the ground truths provided by the experts as a reference for generating the bounding box using LabelImg [21], where the coordinates were stored in Extended Markup Language (XML) files in the PASCAL VOC format, as shown in Figure 4.

#### 3.3.5. Data Partition

Our dataset of 190 patients consisted of 1558 annotated slices. We used the conventional data partition method to train the model, setting the training-to-testing ratio to 80:20. We used slice-wise separation, where 1245 slices were used for training, and 313 slices were used for testing. To combat overfitting, we employed five-fold cross-validation, wherein each fold contained an approximately equal number of randomly sampled slices. Finally, the average of all rounds of training and testing was used to evaluate the model’s performance and establish a derived model.

### 3.4. Deep Learning-Based Localization Model

In this work, we have proposed a CRC localization model that uses a regression-based deep learning approach. Our primary intention was to select a few popular detection models and compare their performances using our CCT image data. Therefore, we used the RetinaNet [22] object localization model with VGG19 as a backbone due to its popularity and better performance with the CT data [23]. Our input consisted of preprocessed images and corresponding XML files with a labeled bounding box’s coordinate information. In addition, we used YOLOv3 with Darknet-53 as the backbone and YOLOv8 [24]. Since YOLO variants are popular for object detection, we compared the performance of RetinaNet with that of 2 YOLO variants: YOLOv3 and the latest model YOLOv8 [25].

### 3.5. Experimental Setup

Our localization models, RetinaNet and YOLOv3, were trained using Tensorflow 1.14 [26], computed on a Linux OS GPU with the following specifications: TITAN RTX 24 GB 4 (Nvidia, Santa Clara, CA, USA), Intel Xeon Scalable Processors (Intel, Santa Clara, CA, USA), 3 UPI up to 10.4 GT/s, with 256 GB memory, and Nvidia-smi 430.40 in the Ubuntu 18.04.3 platform. The other libraries used were Keras 2.1.6, python 3.6.9, numpy 1.18.4, matplotlib 3.2.1, OpenCV 4.1, pillow 7.1.2, and scikit-learn 0.21.3. In addition, we used pytorch 1.8.1+cu101 [27] and torchvision 0.9.1+cu101 for implementing YOLOv8. The hyperparameters used in the models are given as follows in Table 1.

### 3.6. Evaluation Metrics

The IoU was used to evaluate the performance of our localization models, which is calculated via considering the intersection between the predicted bounding box coordinates and the ground-truth bounding box, divided by their union. Using the ground truths acquired from the annotations by the radiologists, we categorized the predictions of the detection networks as true-positive (TP), false-positive (FP), true-negative (TN), and false-negative (FN). We denoted positive as detecting the FLLs and negative as detecting the background. True and false corresponded to the presence of a match and the absence of a match with the annotation results, respectively. We also used different performance metrics such as recall (sensitivity), precision, accuracy, F1 score [28], and average precision (AP) [29], which is calculated as follows:(1)$$\frac{\mathrm{Recall}}{\mathrm{Sensitivity}}=\frac{\mathrm{TP}}{\mathrm{TP}+\mathrm{FN}},$$
(2)$$\mathrm{Precison}=\frac{\mathrm{TP}}{\mathrm{TP}+\mathrm{FP}}\;,\;$$
(3)$$\mathrm{Accuracy}=\frac{\mathrm{TP}+\mathrm{TN}}{\mathrm{TP}+\mathrm{FP}+\mathrm{TN}+\mathrm{FN}}\;,\;$$
(4)$$F1-\mathrm{score}=\frac{2\times \mathrm{Precision}\times \mathrm{Recall}}{\mathrm{Precision}+\mathrm{Recall}},$$

In this work, we performed the localization of only one class, CRC; therefore, the average precision was the same as the precision in our study.

## 4. Results

### 4.1. Slice-Wise Testing

When evaluating the model’s performance with slice-wise analysis, if the model localized a lesion in more than one region in a slice, the lesion with the highest confidence score was considered a correct detection. In such a case, the detection of a lesion with less confidence was discarded instead of considering it a false positive. Conversely, if the lesion had a higher confidence than CRC but was not a lesion, we considered it a false positive. Moreover, if there was no detection in the slices, we considered it a false negative. Table 2 shows the average values of different performance metrics with standard deviation (±) for RetinaNet, YOLOv3, and YOLOv8.

Based on Table 2, it is observed that RetinaNet outperformed YOLOv3 with a precision of 99%. However, YOLOv8 showed the best performance with 96% sensitivity and a 97% F1 score. The model could detect most of the CRC lesions in CT images. Moreover, Figure 5 shows a comparison of a few outputs produced by all three models with the provided ground truths. It can be observed that YOLOv8 has better localization accuracy with better confidence.

Furthermore, Figure 6 shows the performance of the best-performing model YOLOv8 with different curves. The precision–recall curve shows that the model achieved a mAP of 0.984 and an F1 score of 0.97 at a 0.5 IoU threshold. The smooth curves show the stable performance of the model.

### 4.2. Patient-Wise Testing

In addition to performing slice-wise testing, we used 42 patients’ data collected separately to perform patient-wise evaluation. The slices obtained from this set were preprocessed using the methods employed to preprocess the previously collected 190 patients’ data. After preprocessing, all the patients’ slices were fed to the derived localization model, and the outputs produced by the AI-based models were later verified by the experts. In our analysis, if the model could detect a lesion in 70% of a patient’s total positive slices, we considered the patient had a correct detection of CRC. Based on the observations, in our RetinaNet model, 67% of patients (28 out of 42) had correct localizations in more than 70% of slices. YOLOv3 performed similarly to RetinaNet with 67% of patients having correct localizations in more than 70% of slices. However, there were higher false positives (wrong localizations) found in the case of YOLOv3. Meanwhile, 31 patients had correct localizations in more than 70% of slices in the case of YOLOv8, with fewer false positives in comparison with the previous models. The models’ overall performances are recorded in Table 3, where a true positive represents a localization in the correct place, a true negative represents the number of slices with no CRC, FP represents the localization of CRC in either the wrong slice or location, and FN represents missing CRC in any slice. We considered a patient’s sensitivity as 1.00 if the proportion of slices with correct localization was 70% or higher.

Despite the tortuous structure of the colon, our proposed model—YOLOv8 with hyperparameter tuning—could detect CRC with very few false positives. Some outputs produced for one patient are shown in Figure 7. In addition, although some benign lesions were wrongly detected as CRC, as shown in Figure 7 in the second-row third image, the radiologists could focus on such lesions to examine the bowel wall thickening pattern and discard the lesion as CRC. In this way, the derived model can assist radiologists in the early diagnosis of CRC. In addition, clinicians can ask such patients to have follow-ups at regular intervals. With the assistance of the derived model, the number of CRC cases and morbidity due to late detection can be decreased.

## 5. Discussion

This study focused on the development and evaluation of a deep learning-based model for CRC localization in CT scan images. We used different popular object localization models such as RetinaNet and YOLO, which are well known for their suitability for using CT images to detect different organs and associated diseases. Our best-performing model, YOLOv8, exhibited a sensitivity of 96% when five-fold cross-validation-based random slice-wise testing was performed. In addition, some false positives were encountered, the causes of which were analyzed.

Identifying CRC can be challenging in CT scans due to several conditions. The first is the relatively limited attention given to the colon and rectum in studies primarily focused on other diseases. The second is CRC might be ignored if the initial reviewer of these CT slices is not a specialist in reading colorectal lesions, and the patient might be referred to the incorrect department for treatment. In the patient-wise test in our results, 67% of patients (28 out of 42 patients) had correct localizations in more than 70% of slices with RetinaNet. YOLOv3 performed similarly to RetinaNet with 67% of patients having correct localizations in more than 70% of slices. Meanwhile, 31 patients had correct localizations in more than 70% of slices in the case of YOLOv8. In a CT set composed of 100+ slices, a tumor may consist of 3-10 slices. With a detection rate of more than 50% in these slices in a patient, the tumor would be detected by the physician, and CRC may be diagnosed with the model’s assistance. The specificity of these models was 92% for RetinaNet and 96% for YOLOv8. The high specificity of these models can also help physicians to read abdominal CT scans more precisely in CRC detection.

In our study, the colorectal cancer sites were annotated by a radiologist and colorectal surgeon on the CT slices. In addition, the colorectal cancer sites were confirmed by a second radiologist’s reports, the patients’ colonoscopy reports, and the final tumor sites from the operation records if available. The annotation inputs in the deep learning-based model were convincing due to multiple confirmations. In our annotation method, we applied a rectangle to label a tumor rather than contouring the tumor’s border in each slice. This was because colorectal cancer in a CT scan image may not only present wall thickening and contrast uptake but also pericolic change, including inflammation, direct invasion, or lymph node appearance. In addition, labeling a lesion with a rectangle is more efficient and time-saving for physicians responsible for labeling. Pan et al. segmented tumors for invasion evaluation by two clinicians with the ROI and the dilated region of interest (dROI) to input the data into a deep learning model and achieved an AUC value of 0.947 in the T4 and non-T4 (NT4) classification [30]. Huang et al. performed tumor segmentation via manually delineating the ROI along the outline of the visible tumor in the largest three continuous cross-sectional images using 3D Slicer software (version 4.3), and the radiomics signature demonstrated a discriminative performance for high-grade and low-grade CRC, with an AUC of 0.812 in the training dataset and 0.735 in the validation dataset [31]. Another work [32] proposed the efficient segmentation of colorectal tumors in 3D. The work used 3D RoI-aware U-Net for localization and segmentation in magnetic resonance (MR) images. Nonetheless, as only 64 patients were considered in the study, obtaining a large number of MR images was difficult since most patients underwent CT examination. Due to the limited data, the model could suffer from overfitting. Another similar work [33] performed colorectal tumor detection in CT images considering contrary networks. Although the work used a large dataset, it achieved a sensitivity of only 84.73%, and the IoU was 0.56. In addition, the training performances were not discussed.

The precision rate of these models was 40% in RetinaNet and 40% in YOLOv8, and the F-score was 51% in RetinaNet and 64% in YOLOv8. The models’ performances were worse than that of a well-trained physician with a colorectal specialty. The human recognized CRC in CT scans by reading the slices continuously and reconstructing the gut’s tube structure to determine the CRC in the colorectal region. However, deep learning-based models may not work like a human because of the input methods. With a single slice with annotation, a deep learning model cannot build the inference of adjacent slices to more precisely detect the CRC. To improve a model’s performance, more CRC annotations should be featured, and the learning process of the deep learning model should be adjusted.

False positives were encountered during our observations of outputs. When patients’ slices were considered randomly in the model derivation, an average of eight false positives per fold were obtained. This could be due to similar slices in both the training and testing sets. In addition, due to the condition that if multiple lesions were detected in a slice, the lesion with the highest confidence was considered as “CRC”, the remaining lesions were ignored. However, during our evaluation with a new dataset from 42 patients, the number of false positives was higher, which accounted for the reduction in the model’s performance. In addition, there were some cases of false negatives. The primary reason for not detecting CRC lesions was due to size. The lesions with smaller sizes (less than 2.5 cm) were missed by our model, which may have resulted in missing our primary goal of the early detection of CRC.

Our current work has some limitations. First, we only used a single institutional dataset, which could affect the model’s performance. Second, we only used single-phase CT for our analysis. We want to consider if there is a requirement for multi-phase CT in CRC detection, such as adding horizontal sections. In addition, more tumor-annotated CT slices should be applied to increase the model’s learning power.

## 6. Conclusions

In this work, we used a single institutional dataset for the model establishment. Initially, we considered 190 patients’ CT scan images for the model derivation, where our proposed model achieved an F1 score of 97% with a mAP of 0.984 in localizing the CRC region in the CCT slices. Furthermore, we tested the robustness of our model with another 42 patients’ data, where the model performed with 96% accuracy. We plan to perform an ablation study considering customized detection models in future work with multi-institutional images and the clinical data of CRC patients.

## Figures and Tables

**Figure 1 bioengineering-10-00972-f001:**
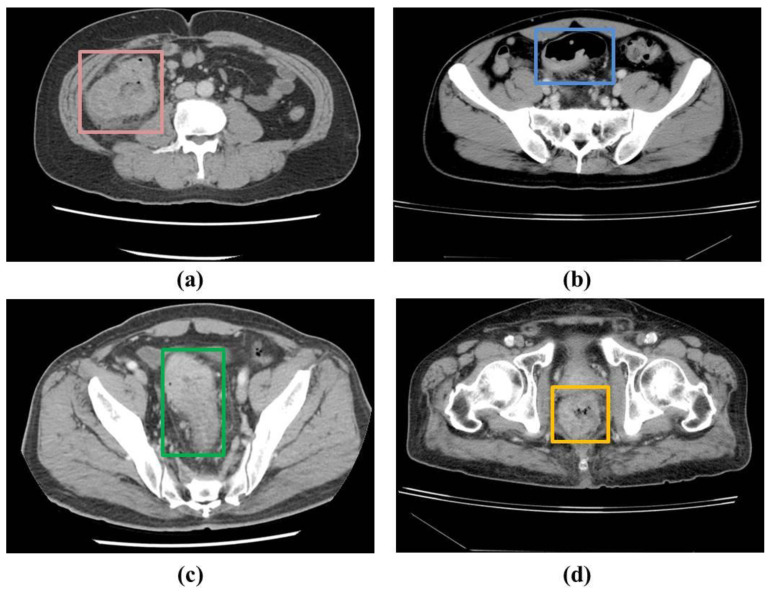
Examples of the ground truths provided by the experts: (**a**,**b**) colon region, and (**c**,**d**) rectum region.

**Figure 2 bioengineering-10-00972-f002:**
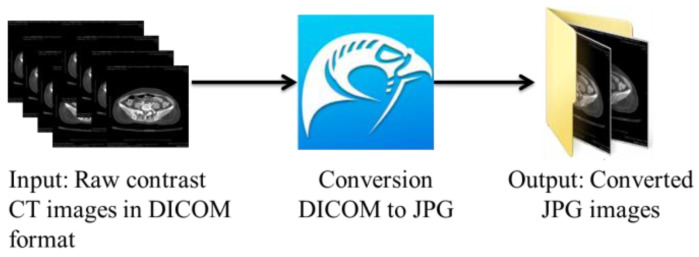
Conversion of DICOM to JPG.

**Figure 3 bioengineering-10-00972-f003:**
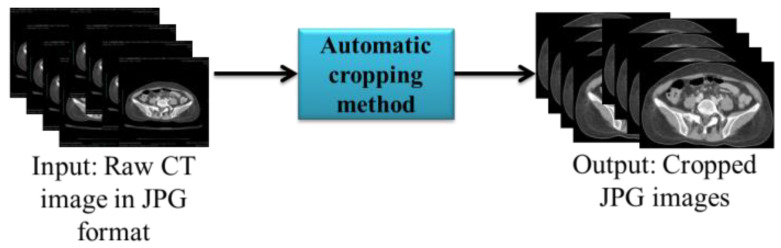
Automatic cropping of slices.

**Figure 4 bioengineering-10-00972-f004:**
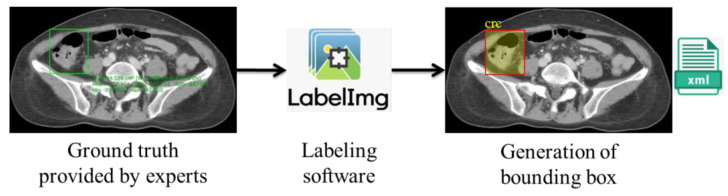
Generation of bounding box.

**Figure 5 bioengineering-10-00972-f005:**
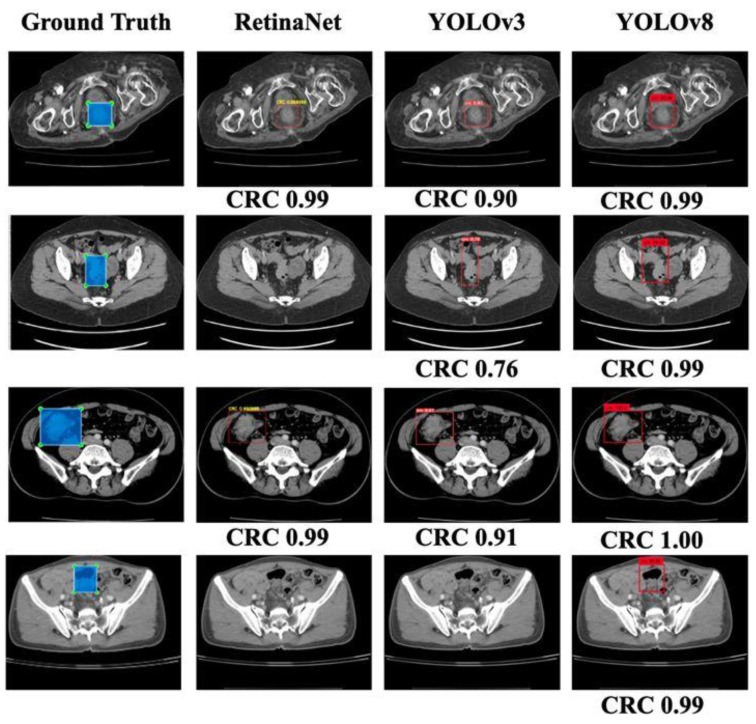
Slice-wise testing showing CRC lesions localized via different models with confident scores, where the first column represents the ground truth, and second–fourth columns show the bounding box for the localized CRC region as predicted by the different models. The absence of the bounding box shows the inability of the model in localization.

**Figure 6 bioengineering-10-00972-f006:**
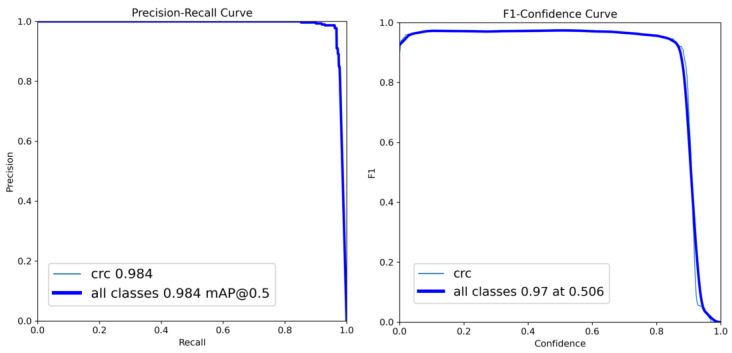
Precision–recall curve and F1-confidence curve for YOLOv8: best performing model.

**Figure 7 bioengineering-10-00972-f007:**
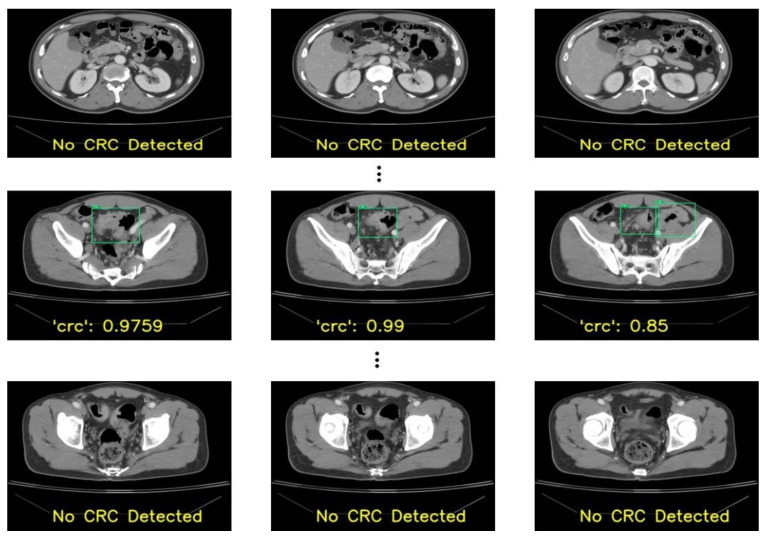
Patient-wise testing showing CRC lesions localized (green box) with confident scores that were verified as the correct localization.

**Table 1 bioengineering-10-00972-t001:** List of hyperparameters used for each model.

Localization Model	RetinaNet	YOLOv3	YOLOv8
Backbone	VGG19	Darknet-53	CSPDarknet-53
Batch size	2	2	2
Learning rate	0.0001	0.0001	0.001
Activation function	ReLU	Leaky ReLU	Mish
Optimizer	Adam	Adam	Adam
# of epochs	50	50	200

**Table 2 bioengineering-10-00972-t002:** Performance metrics with standard deviation (±) of different deep learning models for localizing CRC in CCT images.

Models	Evaluation Metrics			
	Sensitivity	Precision	Accuracy	F1−Score
RetinaNet	0.94 (±0.007)	0.99 (±0.007)	0.93 (±0.011)	0.97 (±0.005)
YOLOv3	0.94 (±0.017)	0.98 (±0.011)	0.92 (±0.021)	0.96 (±0.011)
YOLOv8	0.96 (±0.003)	0.99 (±0.002)	0.95 (±0.005)	0.97 (±0.002)

**Table 3 bioengineering-10-00972-t003:** Performance metrics with standard deviation (±) of different deep learning models in localizing CRC in CT images in unseen 42 patients’ data.

Model	Evaluation Metrics				
	Recall	Precision	Specificity	Accuracy	F1 Score
RetinaNet	0.74 (±0.39)	0.40 (±0.24)	0.94 (±0.03)	0.92 (±0.03)	0.51 (±0.28)
YOLOv3	0.76 (±0.36)	0.27 (±0.16)	0.87(±0.04)	0.87 (±0.04)	0.39 (±0.21)
YOLOv8	0.83 (±0.29)	0.60 (±0.22)	0.97(±0.01)	0.96 (±0.01)	0.64 (±0.31)

## Data Availability

Data is unavailable due to the privacy of patients.

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
