# Peer review of "Localization of Colorectal Cancer Lesions in Contrast-Computed Tomography Images via a Deep Learning Approach"

_bioengineering, 2023, doi:10.3390/bioengineering10080972_

Round 1
Reviewer 1 Report
The authors utilized abdominal computed tomography images and YOLOV8 deep learning neural network to achieve more accurate localization of colorectal cancer lesions. This is a very meaningful job. There are several issues that need to be noted to make the paper better.
Firstly, the font size of formulas 1 to 4 should be consistent and the layout should be neat.
Secondly,the authors can provide a picture that includes the Technology roadmap of this paper.
Thirdly, from the data perspective, YOLOV8 can greatly improve the performance of the model. Can the authors conduct ablation experimental analysis?
Author Response
We would like to express our sincere gratitude to reviewer for giving us the opportunity to revise our work and for the valuable comments and constructive suggestions. Based on the comments, we have revised the manuscript thoroughly. Hope this revised version would be acceptable.
Comment: Firstly, the font size of formulas 1 to 4 should be consistent and the layout should be neat.
Response: The font size and layout of formulas are modified to maintain consistency.
Comment: Secondly, the authors can provide a picture that includes the Technology roadmap of this paper.
Response: Thank you for the suggestion. The picture that includes the technology roadmap of this work are provided in the form of graphical abstract (Page # 2).
Comment: Thirdly, from the data perspective, YOLOV8 can greatly improve the performance of the model. Can the authors conduct ablation experimental analysis?
Response: In this work, we have compared the performance of popular models RetinaNet, YOLOv3 and YOLOv8. Currently, we have used pretrained models and have performed extensive hyperparameter tuning for model comparisons. Thank you for the valuable suggestion. We will consider the customized model in our future work to perform the ablation study, which is included in the conclusion as shown in Page #13.

Reviewer 2 Report
This work proposed the use of three deep learning algorithms (RetinaNet, YoLoV3 and YoLoV8) for localization of colorectal cancer in CT image and achieved 0.97 (±0.002) F1-score, mAP 0.984 when performed slice-wise testing and achieved 0.83 (±0.29) sensitivity, 0.97 (±0.01) specificity and 0.96 35(±0.01) accuracy. It’s interesting. The major comments are as below:
1. The abstract should be concise. There are too many background introduction.
2. Line 288-289: equation of precision and accuracy should be aligned in a line.
3. Tables should be presented in a three-line pattern.
4. Figure 5: It seems that the sensitivity of RetinaNet is worse than YOLOv3. Why?
This work proposed the use of three deep learning algorithms (RetinaNet, YoLoV3 and YoLoV8) for localization of colorectal cancer in CT image and achieved 0.97 (±0.002) F1-score, mAP 0.984 when performed slice-wise testing and achieved 0.83 (±0.29) sensitivity, 0.97 (±0.01) specificity and 0.96 35(±0.01) accuracy. It’s interesting. The major comments are as below:
1. The abstract should be concise. There are too many background introduction.
2. Line 288-289: equation of precision and accuracy should be aligned in a line.
3. Tables should be presented in a three-line pattern.
4. Figure 5: It seems that the sensitivity of RetinaNet is worse than YOLOv3. Why?
Author Response
We would like to express our sincere gratitude to reviewer for giving us the opportunity to revise our work and for the valuable comments and constructive suggestions. Based on the comments, we have revised the manuscript thoroughly. Hope this revised version would be acceptable.
Comment: The abstract should be concise. There are too many background introduction.
Response: Based on the suggestion, the abstract is rewritten concisely in the revised version.
Comment: Line 288-289: equation of precision and accuracy should be aligned in a line.
Response: The equations are aligned in the modified version of the manuscript.
Comment: Tables should be presented in a three-line pattern.
Response: The tables are modified in three-line pattern as per the suggestion.
Comment: Figure 5: It seems that the sensitivity of RetinaNet is worse than YOLOv3. Why?
Response: Figure 5 illustrates few samples of outputs produced by different localization models. Overall, both RetinaNet and YOLOv3 have similar performance 0.94 (Table 2, Page #9) in terms of sensitivity.

Reviewer 3 Report
The work of the authors is interesting. I have only some minor points to improve clarity for a broader readership:
-in the introduction/background part, please add some discussion of the importance of detecting early cancer with CT in comparison with colonoscopy screening programs, whether there are peculiarities in the country of authors, and more referencing on the topic
- heading of the results section should be only "results" because the discussion follows in another different section
- deeper comparison with other works doing similar research is advisable in the discussion section
I always advise to non native English writers (like me) careful reviewing of grammar and typos
Author Response
We would like to express our sincere gratitude to reviewer for giving us the opportunity to revise our work and for the valuable comments and constructive suggestions. Based on the comments, we have revised the manuscript thoroughly. Hope this revised version would be acceptable.
Comment: In the introduction/background part, please add some discussion of the importance of detecting early cancer with CT in comparison with colonoscopy screening programs, whether there are peculiarities in the country of authors, and more referencing on the topic.
Response: The relevance of CT over colonoscopy is added in the introduction part (Page # 3).
Comment: Heading of the results section should be only "results" because the discussion follows in another different section.
Response: The heading is modified.
Comment: Deeper comparison with other works doing similar research is advisable in the discussion section.
Response: Thanks for the suggestion. We have added two relevant works [Reference #: 32,33] in the discussion section (Page #13).